# HtrA1 in Gestational Diabetes Mellitus: A Possible Biomarker?

**DOI:** 10.3390/diagnostics12112705

**Published:** 2022-11-05

**Authors:** Giovanni Tossetta, Sonia Fantone, Rosaria Gesuita, Gian Carlo Di Renzo, Arun Meyyazhagan, Chiara Tersigni, Giovanni Scambia, Nicoletta Di Simone, Daniela Marzioni

**Affiliations:** 1Department of Experimental and Clinical Medicine, Università Politecnica delle Marche, 60126 Ancona, Italy; 2Clinic of Obstetrics and Gynaecology, Department of Clinical Sciences, Università Politecnica delle Marche, Salesi Hospital, Azienda Ospedaliero Universitaria, 60126 Ancona, Italy; 3Centre of Epidemiology and Biostatistics, Università Politecnica delle Marche, 60126 Ancona, Italy; 4Department of Obstetrics, Gynecology and Perinatology, IE Sechenov First State University, 119991 Moscow, Russia; 5Wayne State University Medical School and Perinatal Research Branch, NIH-NICHD, Detroit, MI 48201, USA; 6Fondazione Policlinico Universitario A. Gemelli IRCCS, Dipartimento di Scienze della Salute della Donna, del Bambino e di Sanità Pubblica, 00168 Roma, Italy; 7Istituto di Clinica Ostetrica e Ginecologica, Università Cattolica del Sacro Cuore, 00168 Roma, Italy; 8Department of Biomedical Sciences, Humanitas University, 20072 Milan, Italy; 9IRCCS Humanitas Research Hospital, 20089 Milan, Italy

**Keywords:** HtrA1, GDM, marker, first trimester, pregnancy

## Abstract

Background: The high-temperature requirement A 1 (HtrA1) is a multidomain secretory protein with serine-protease activity, expressed in many tissues, including placenta, where its expression is higher in the first trimester, suggesting an association of this serine protease in early phases of human placenta development. In this study, we evaluated maternal serum HtrA1 levels in the first and third trimester of gestation. In particular, we evaluated a possible role of HtrA1 as an early marker of gestational diabetes mellitus (GDM) in the first trimester of gestation. Methods: We evaluated HtrA1 serum levels in the third trimester (36–40 weeks) in normal pregnancies (n = 20) and GDM pregnancies (n = 20) by using ELISA analysis. Secondly, we performed the same analysis by using the first trimester sera (10–12 weeks) of healthy pregnant women that will develop a normal pregnancy (n = 210) or GDM (n = 28) during pregnancy. Results: We found that HtrA1 serum levels in the third trimester were higher in pregnancies complicated by GDM. Interestingly, higher HtrA1 serum levels were also found in the first trimester in women developing GDM later during the second–third trimester. No significant differences in terms of maternal age and gestational age were found between cases and controls. Women with GDM shown significantly higher pre-pregnancy BMI values compared to controls. Moreover, the probability of GDM occurrence significantly increased with increasing HtrA1 levels and BMI values. The ROC curve showed a good accuracy in predicting GDM, with an AUC of 0.74 (95%CI: 0.64–0.92). Conclusions: These results suggest an important role of HtrA1 as an early predictive marker of GDM in the first trimester of gestation, showing a significative clinical relevance for prevention of this disease.

## 1. Introduction

Gestational diabetes mellitus (GDM) is a gestational complication, the frequency of which is increasing in the world, showing a prevalence among 5–15% of all pregnancies worldwide [1]. GDM is a glucose intolerance occurring in the second and third trimesters of pregnancy that leads to maternal and fetal/neonatal complications [2,3] and it is generally diagnosed at 24–28 weeks of gestation [3,4]. GDM onset is a result of a beta cell dysfunction due to chronic insulin resistance, which has been reported as being an important cause of GDM, type 2 diabetes mellitus (DM) and obesity. Maternal obesity, physical inactivity, increased maternal age and GMD history are known risk factors for GDM [5]. Occurrence of GDM can lead to severe complications for both mother and child. In fact, neonates from patients with GDM can develop many complications, including macrosomia, neonatal hypoglycemia and hyperbilirubinemia, but also a higher risk of obesity, insulin resistance and type 2 diabetes in adulthood. Mothers with GDM are at risk of caesarean delivery, hypertension, preeclampsia (PE) and higher incidence of type 2 diabetes mellitus (T2DM) and cardiovascular disease incidence later on [6]. Moreover, diabetes leads to important renal complications in adults and young people that can interest mothers affected by gestational diabetes and their children [7,8,9,10]. Thus, an early diagnosis and treatment of GDM is very important to prevent possible complications of this pregnancy disorder. It has been suggested that chronic low-level inflammation is one of the most important causes of insulin resistance onset, since chronic exposure to pro-inflammatory cytokines can impair insulin signaling pathways and decrease insulin secretion from beta cells [5]. Inflammation is the result of cellular damage and characterizes many pathologies, including cancer [11,12,13] and immune diseases [14], but also pregnancy complications, such as preeclampsia and preterm birth [15,16,17,18]. Chronic low-grade inflammation is a pathological condition, playing an important role in the pathogenesis and progression of some pathologies, including GDM, metabolic syndrome, DM and cardiovascular diseases [19,20]. The first trimester of pregnancy is a window of opportunity for early identification, prediction and treatment for many pregnancy complications. Since GDM has important consequences for both the pregnant woman and the fetus, it is important to obtain an early identification of GDM, especially for the detection of high-risk pregnancies in order to define their proper follow-up, reducing morbidity in mother and fetus. 

The high-temperature requirement A (HtrA) family protein includes HtrA1 (L56 or PRSS11), HtrA2/Omi, HtrA3 (PRSP) and HtrA4. HtrA1 is a secreted serine protease characterized by the presence of five domains. Starting from the N-terminus, it contains a signal sequence for secretion (SP), an IGFBP/mac25-like domain (IGFBP), a kazal-type inhibitor domain (KI), a proteolytic domain (PD) and one PDZ domain at C-terminus [21]. HtrA1 is the most studied member of the HtrA family proteins and plays a key role in regulating many cell processes, including cell proliferation and differentiation, being an important regulator of the physiological development of many organs, including placenta [11,22,23,24,25,26,27]. In fact, it has been reported that HtrA1 is expressed in the placenta during gestation and its expression increases from the first to the third trimester of normal pregnancy [22]. GDM placentas show histological alterations associated with villous immaturity, such as increased cytotrophoblast cells and a higher number of branches per capillary in terminal villi (known as hypervascularity) [28]. In previous studies, we found that these placental alterations are associated with altered HtrA1 maternal levels in patients with PE [29] and spontaneous preterm birth (sPTB) [30]. Moreover, HtrA1 expression was found to be altered also in placentas from gestational trophoblastic diseases [31]. Importantly, it has been reported that HtrA1 plays a main role in placental vascular tree development, as demonstrated in HtrA1 knockout mice [32]. Thus, possible alterations in HtrA1 during pregnancy could significantly impair pregnancy outcome. 

The aim of this study was to determine whether HtrA1 plasma concentration at the first trimester of gestation is altered in pregnant women that will develop GDM later. Moreover, we aimed to evaluate a possible role of HtrA1 as an early marker of GDM in the first trimester of pregnancy, identifying women at risk for developing GDM in the second or third trimester.

## 2. Materials and Methods

### 2.1. Study Population and Exclusion Criteria

For the analysis of HtrA1 levels in third trimester of gestation in normal pregnancies and pregnancy complicated by gestational diabetes we conducted a retrospective case–control study. For the analysis of HtrA1 levels in the first trimester of gestation we performed an observational prospective cohort study of healthy pregnant women at 10–12 weeks of gestation. Blood samples were collected from pregnant women attending antenatal care visit at Department of Obstetrics and Gynecology of the Università Politecnica delle Marche (Ancona, Italy) and at Department of Woman and Child Health, Fondazione Policlinico A. Gemelli IRCCS (Rome, Italy), prospectively followed up until delivery. All blood draws used for the experiment were performed under fasting conditions in the morning. The study was performed following the Declaration of Helsinki and approved by the Institutional Ethics Committee of the Università Politecnica delle Marche (Study protocol 2019/172) and Fondazione Policlinico A. Gemelli IRCCS (Study Code 20345/22). All women recruited in this study provided written informed consent. 

The blood samples were collected from two cohorts: third trimester (31–37 weeks) group (GDM, n = 20; CTRL, n = 20) and first trimester (10–12 weeks) group (GDM, n = 28; CTRL, n = 210). The diagnosis of GDM in pregnancy was performed by Oral Glucose Tolerance Test (OGTT) between the 24th and the 28th weeks of gestation and GDM was diagnosed according to the guidelines of the International Association of Diabetes and Pregnancy Study Groups (IADPSG) [33].

Baseline demographics, medical history including obstetric history, current and before-pregnancy habits (smoking, eating, physical activity) were collected through an interview and anthropometric characteristics were taken (Body Mass Index—BMI) for pregnant women attending a routine antenatal care visit. 

Stillbirths and presence of chromosomal and other fetal anomalies were excluded. A history of hypertension, renal disease, cardiac disease, diabetes mellitus, thyroid and immunologic diseases and congenital or acquired thrombophilia were used as exclusion criteria for the study.

### 2.2. Plasma Collection and HtrA1 ELISA

Blood samples were obtained by Vacutainer™ venipuncture of the median cubital vein after overnight fasting. Plasma samples were prepared from fresh EDTA venous blood centrifuged at 1500× *g* for 15 min at 4 °C. Plasma was then aliquoted and stored at −80 °C until use. Plasma HtrA1 concentration was measured by using commercial ELISA kit (#MBS2886125: MyBioSource, Inc., San Diego, CA, USA). The minimum detectable dose of HtrA1 is less than 20 pg/mL as described in the instruction manual of ELISA kit. In addition, intra-assay and inter-assay CV% are ≤10% and ≤6.3%, respectively. The measurements were conducted in duplicate, according to the manufacturer’s recommended protocol. One hundred microliters of plasma samples was used for each well. Internal negative and positive quality controls were supplied with the kit.

### 2.3. Statistical Analysis

A non-parametric approach to descriptive analysis was followed since quantitative variables were not normally distributed. The variables were summarized using absolute and percentage frequencies (qualitative variables) or median and interquartile range (IQR, quantitative variables). Comparisons between healthy women and those with GDM were performed using Chi-square test or Wilcoxon rank sum test. 

Multiple logistic regression analysis was used to estimate the association between the development of GDM after the first trimester of gestation (dependent dichotomous variable) and women’s clinical characteristics evaluated at the first trimester. Likelihood ratio (LR) test and Hosmer–Lemeshow (HL) test were used to select the most parsimonious model and to evaluate the model’s goodness of fit. A Receiver Operating Curve was estimated to assess the ability of HtrA1 to predict GDM, adjusted for the clinical features significantly associated with GDM in the logistic model.

The R programming language and environment (https://www.r-project.org/about.html; access date 15 September 2022) was used for the statistical analyses and a probability of 5% was set for statistical significance.

## 3. Results

### 3.1. HtrA1 Plasma Concentrations Are Increased in Third Trimester of Pregnancy with GDM

The first aim of the study was to assess HtrA1 plasma concentration at the third trimester of pregnancy complicated by GDM compared to healthy control pregnancies. The demographic characteristics of patients with normal pregnancy (n = 20) and pregnancy complicated by GDM (n = 20) included in the study are shown in Table 1. The median age of patients in the GDM group was 36 years (interquartile range, IQR: 35–39 years) while in the control group was 32 years (IQR: 30–36 years), highlighting a statistically significant increase in maternal age of patients with GDM (*p* = 0.006). The median gestational age at delivery in the GDM group was 39 weeks (IQR: 39–40 weeks) while in the control group was 40 weeks (IQR: 38–41 weeks), showing no differences in gestational age at delivery between GDM and normal pregnancies. Moreover, the median birth weight of children of women with GDM was 3.2 Kg (IQR: 2.8;3.4) while for normal pregnancies was 3.5 Kg (IQR: 3.3;3.6), proving that children of women with GDM had a statistically significant lower birth weight (*p* = 0.041). This result can be explained by the data reported in the literature, showing that 7% of newborns from GDM mothers are small for gestational age (SGA) [34]. 

Interestingly, we found that the median of HtrA1 concentration of GDM pregnancies was 4.3 ng/mL (IQR: 3.4;6.0) while in normal pregnancies was 1.9 ng/mL (IQR: 0.8;4.0), showing that pregnancy complicated by GDM had a statistically significant higher concentrations of HtrA1 compared to healthy pregnancies (*p* = 0.004), proving that HtrA1 is altered in pregnancy complicated by GDM. The individual values of HtrA1 and clinical characteristics of patients are available in Appendix A. 

### 3.2. HtrA1 Could Be an Early Marker of GDM in First Trimester of Pregnancy

Since HtrA1 plasma concentration was increased in women with GDM at the third trimester of pregnancies, we wondered whether HtrA1 plasma concentration could be increased also during the first trimester of pregnancy in women (healthy at moment of sampling) that will develop GDM later in pregnancy. To this aim, we evaluated HtrA1 plasma concentrations in 238 women attending a routine antenatal care visit at the first trimester of pregnancy. Among them, 28 women were diagnosed with GDM at the moment of OGTT (24–28 weeks of gestation). As shown in Figure 1, there were no significant differences in maternal age and gestational age between the GDM (n = 28) and control (n = 210) group but women who would develop GDM showed significantly higher BMI values before pregnancy and increased HtrA1 plasma concentrations in the first trimester of gestation. The individual values of HtrA1 and clinical characteristics of patients are available in Appendix A.

Moreover, there were no significant differences in maternal smoking, eating habits (low/poor nutrition) and physical activity during pregnancy (see Table 2). 

Interestingly, we found that the probability of developing GDM significantly increased with increasing HtrA1 and BMI values (see Table 3). In particular, it increases 2.5-times and of 14% for every added unit of HtrA1 and BMI, respectively. Moreover, we found an increased risk of 43% for developing GDM in the presence of poor nutrition. 

The ROC curve for the predictiveness of GDM showed a good level of accuracy with an AUC of 0.74 (95%CI: 0.64–0.92) (Figure 2).

## 4. Discussion

It has been previously demonstrated that HtrA1 is localized in the placenta, playing a key role in extracellular matrix remodeling [26] and vasculogenesis [32]. In particular, it is mainly expressed in the villous cytotrophoblast in the first trimester and in the syncytiotrophoblast in the third trimester of gestation [31]. Since HtrA1 plasma levels increased from the first to the third trimester of gestation during normal pregnancy [35], we can assume that the placenta is a source of circulating HtrA1, but other sources, such as maternal vessels, may significantly contribute to HtrA1 release in the blood stream. We demonstrated for the first time that HtrA1 was significantly increased in the serum of patients with GDM at the third trimester of pregnancy compared to normal control pregnancy, letting us think that HtrA1 could be altered already in the first trimester. Our prospective study is the first that investigated HtrA1 maternal plasma levels in pregnant women at the first trimester of gestation that will develop GDM later, in order to evaluate the possible role of HtrA1 in GDM prediction. The key finding of this study is that HtrA1 levels are significantly higher in the GDM group than in the healthy control group. Our model had good accuracy in predicting GDM, showing an ROC curve with an AUC equal to 0.74. Moreover, we found that a high BMI before pregnancy significantly increases the risk for developing GDM. These results were tightly correlated with the nutrition habits of patients, since we found an increased risk of 43% for developing GDM in the presence of poor nutrition (intended as not healthy). In fact, poor nutrition (e.g., rich in fat and carbohydrates) can significantly increase patient BMI. 

These data were in agreement with previous studies, showing an increased risk of developing GDM in the presence of a high BMI before pregnancy [36,37]. It is possible to hypothesize that the placenta responds to altered glucose metabolism already in the first trimester, secreting high HtrA1 levels, although GDM will only be diagnosed later in pregnancy when the altered maternal blood glucose levels become visible at the OGTT test. However, we cannot state if the abnormal increase in HtrA1 in the first trimester of pregnancies can be a placental stress response to altered glucose metabolism, as we hypothesized, or one of the causes of GDM onset, such as higher maternal BMI and maternal adipose tissue inflammation that leads to insulin resistance [38].

High blood glucose levels (hyperglycemia) found in diabetes lead to chronic low-grade inflammation, bringing microvascular complications that can worsen in cardiovascular diseases, contributing to the morbidity and mortality associated with diabetes [39]. Due to this systemic alteration in the presence of hyperglycemia, it is possible to hypothesize that the high serum levels of HtrA1 found in pregnancies complicated by GDM are due to the chronic inflammation that characterizes this pathology, leading to both maternal vascular damage (since HtrA1 is expressed in the endothelial cells of maternal vessels) and placental (since HtrA1 it is expressed by the syncytiotrophoblast, the outer layer of the trophoblast in contact with the maternal blood present in the intervillous space and, therefore, with the cytokines contained therein).

This hypothesis is further validated by the fact that HtrA1 serum levels are also altered in other inflammatory conditions. In fact, our research group has previously demonstrated that HtrA1 serum levels also increased during the first trimester of gestation in other gestational complications, such as PE [29] and sPTB [30]. Thus, HtrA1 quantification in first-trimester plasma levels may predict the occurrence of different gestational complications, including GDM. However, since HtrA1 increases in all three of the pregnancy complications mentioned above, it is not a specific marker of GDM prediction if used alone. In fact, since PE, sPTB and GDM share an inflammatory base in their pathogenesis, we can suggest that HtrA1 may act more as an inflammatory marker rather than a GDM marker. Thus, HtrA1 may predict an inflammation status in healthy women at the first trimester of gestation but, if used with other first-trimester GDM markers, such as soluble CD163 (sCD163) [40], and associated with pre-pregnancy BMI, it may significantly increase the GDM prediction rate at the first trimester. Therefore, further studies could be focused on evaluating the role of HtrA1 in combination with other markers in predicting GDM in the first trimester of pregnancy.

Our study showed important clinical implications, because first-trimester prediction of the risk for GDM development (or other complications), could promote earlier lifestyle changes and, if necessary, initiate pharmacological treatments at an earlier pregnancy stage in order to improve pregnancy outcomes. 

The prospective design of the study is the major strength because it allowed us to evaluate the HtrA1 levels when no complications of pregnancy are present. Moreover, since patients were followed until delivery, we could evaluate the impact of the clinical and demographic characteristics of patients in GDM onset. Our study also presents some limitations, such as the small sample size at the first and third trimester. However, we used the data at the third trimester only to evaluate possible changes in HtrA1 expression in GDM. Moreover, the prospective study at the first trimester showed good accuracy in GDM prediction and confirmed the data reported in the literature, showing an increase in HtrA1 expression from the first to the third trimester of gestation in normal pregnancy [22]. 

In conclusion, in this study, we demonstrated that evaluating HtrA1 maternal plasma levels in the first trimester of pregnancy could allow for an early identification of pregnancy at risk of developing gestational complications, such as GDM, later in pregnancy. These findings could have strong clinical relevance in terms of disease prevention. These pregnant women may be on a controlled diet right away and may be treated appropriately in order to prevent or delay GDM onset, improving maternal and fetal outcomes and reducing adverse cardiovascular and metabolic events in the mother after pregnancy, as well as reducing the economic and healthcare impacts caused by this pathology. 

Moreover, since HtrA1 levels can be tested by ELISA, a simple and low-cost assay, measuring HtrA1 levels, could be easily included in clinical practice in order to verify if pregnant women with high HtrA1 levels at the first trimester of gestation and subjected to a controlled diet could avoid GDM onset. 

## Figures and Tables

**Figure 1 diagnostics-12-02705-f001:**
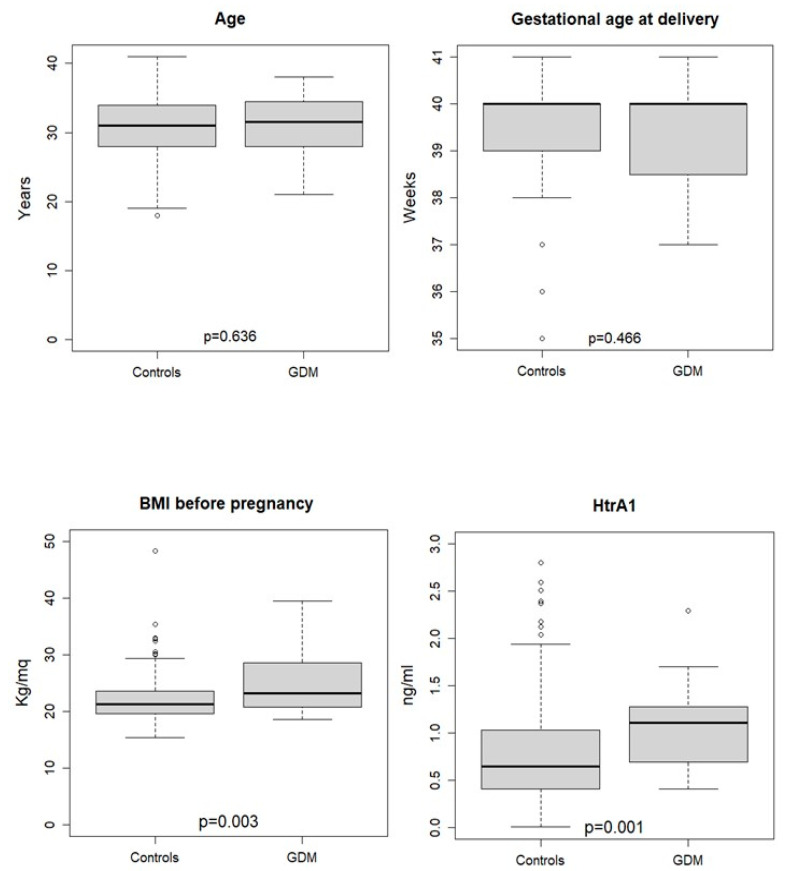
Women’s demographic and clinical characteristics according to health status at the first trimester of gestation. *p*-values refer to Wilcoxon rank sum test.

**Figure 2 diagnostics-12-02705-f002:**
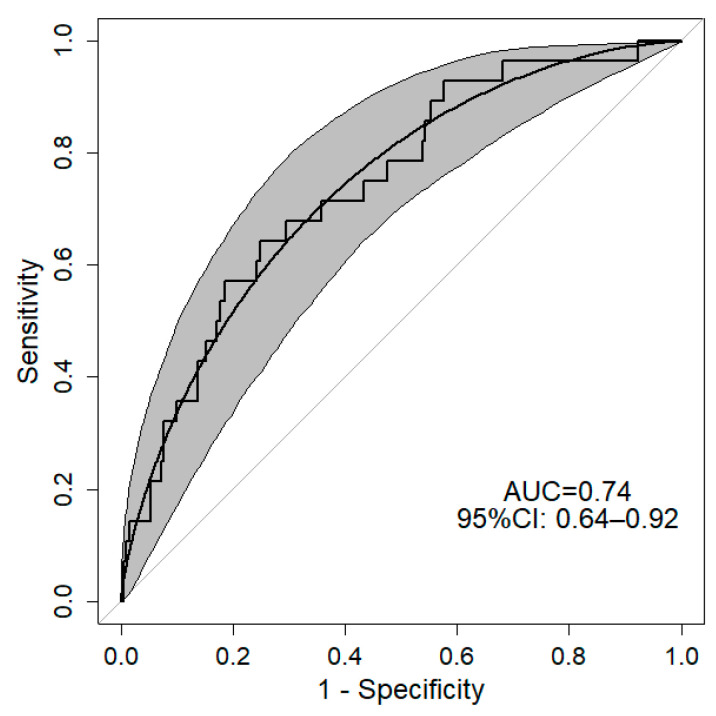
Observed and smoothed ROC curves with 95% confidence interval for the predictiveness of GDM. Grey area identifies 95% confidence interval of ROC. AUC: Area Under Curve; 95%CI: 95% Confidence Interval. ROC was estimated as a function of HtrA1 type of nutrition in the third trimester of gestation and BMI before pregnancy.

**Table 1 diagnostics-12-02705-t001:** Demographic and clinical characteristics according to health status at delivery.

	Healthy Pregnancies (n = 20)	Pregnancies with GDM (n = 20)	*p*
Maternal age (years)	32 (30;36)	36 (35;39)	**0.006**
BMI (Kg/m^2^)	24 (22;25)	21 (21;25)	0.229
Gestational age at delivery (Weeks)	40 (38;41)	39 (39;40)	0.107
Gestational Age at sampling (Weeks)	34 (31;36)	36 (32;37)	0.124
HtrA1 (ng/mL)	1.9 (0.8;4.0)	4.3 (3.4;6.0)	**0.004**
Birth weight (Kg)	3.5 (3.3;3.6)	3.2 (2.8;3.4)	**0.041**

Values are medians and IQR; *p*-values refer to Wilcoxon rank sum test. Values are absolute frequencies and percentages; *p*-values refer to Chi-square test. BMI: Body Mass Index.

**Table 2 diagnostics-12-02705-t002:** Women’s lifestyle according to health status at the first trimester of gestation.

	Healthy Pregnancies (n = 210)	Pregnancies with GDM (n = 28)	*p*
Smoking (yes)	55 (26.2%)	7 (25.0%)	0.999
Low/poor nutrition	137 (65.2%)	14 (50.0%)	0.173
Physical activity (yes)	140 (66.7%)	19 (67.9%)	0.999

Values are absolute frequencies and percentages; *p*-values refer to Chi-square test.

**Table 3 diagnostics-12-02705-t003:** Factors evaluated at the first trimester of gestation associated with GDM. Results of logistic regression analysis.

	OR	95% CI	*p*
HtrA1 (ng/mL)	2.50	1.2; 5.16	**0.013**
Maternal age (years)	1.02	0.93; 1.13	0.628
BMI before pregnancy (kg/m^2^)	1.14	1.05; 1.25	**0.002**
Gestational age (weeks)	0.90	0.63; 1.29	0.540
Nutrition (good vs. poor/low)	0.57	0.34; 0.95	**0.036**
Smoke (yes vs. no)	0.91	0.33; 2.29	0.849
Physical activity (yes vs. no)	1.42	0.58; 3.75	0.457

Likelihood ratio (LR) test: χ^2^_7_ = 19.46, *p* = 0.007; HL test: χ^2^_8_ = 5.3, *p* = 0.725.

## Data Availability

The data presented in this study are available on request from the corresponding author.

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
