# Peer review of "HtrA1 in Gestational Diabetes Mellitus: A Possible Biomarker?"

_diagnostics, 2022, doi:10.3390/diagnostics12112705_

Round 1

Reviewer 1 Report

In this manuscript Tossetta and colleagues evaluated the predictive value of HtA1 for the development of gestational diabetes in the first trimester of pregnancy. 

The study is well developed, generally well written and well illustrated. Only some minor implementations may be needed. In particular:

Introduction: Authors should highlight the possible complications that may precede and accompany diabetes. In fact several markers of diabetes and pre-diabetes-induced early kidney damage may be associated with HtA1 and the development of gestational diabetes and its complications. Diabetes leads to important renal complications in adults and young people. This may be the result of a chronic low-grade inflammation that also characterise gestational diabetes. Moreover, these complications can interest mothers affected by gestational diabetes and their children (see PMID:  33814206, 35507146, 34211943, 30063409) and could be avoided with an early diagnosis and treatment of gestational diabetes as suggested by the authors.

Author Response

We implemented the manuscript according to the reviewer suggestion. 

Reviewer 2 Report

In this article, the authors measured serum levels of HtrA1 in pregnant women using an ELISA approach, first in healthy women (n=20) and women with GDM (n=20) in the third trimester of pregnancy, and then in sera from women at the first trimester of pregnancy (n=238) from which 28 developed GDM. Results showed that HtrA1 serum levels were higher in women with GDM in the third trimester of pregnancy, and in women in the first trimester that developed GDM later in pregnancy. In addition, the results showed that the probability of developing GDM increased with increasing levels of HtrA1 and BMI values. These results suggest that HtrA1 could be an early predictive marker for GDM, and thus has clinical relevance.  Nevertheless, the increase of HtrA1 level in serum seems not to be specific for GDM, but also for other pregnancy complications, therefore the title could be misleading and might require a slight modification, for example “HtrA1 as early marker of pregnancy complications including gestational diabetes mellitus”

 Please provide a Table with complete information (baseline demographics -including ethnicity-, clinical history, habits, etc.) for each patient in a supplementary file. This information could be useful for the readers.

 Lines 169-173: “Moreover, the median birth weight of children of women with GDM was 3.2 Kg (IQR: 2.8;3.4) while for normal pregnancies was 3.5 Kg (IQR: 3.3;3.6) proving that children of women with GDM had a statistically significant lower birth weight (P = 0.041), which was possibly due to the lower gestational age at delivery found in pregnancies complicated with GDM”. Some GDM babies are large for gestational age, the authors should report the weight of the baby according to gestational age, otherwise is not comparable to full term babies.

 Even if HtrA1 serum levels were significantly higher in women with GDM, individual variability is high; it would be useful if the authors provide individual HtrA1 values of healthy and GDM pregnancies in a supplementary file.

 The authors discuss major strengths of the study, however they should also discuss limitations.

 Minor points:

Line 63: “… inflammation IS one of the most important CAUSES OF insulin resistance…”

Line 102: “… third trimester of GESTATION in normal…”

Line 126: “A history…”

Line 155: R is a programming language and environment for statistical computing and graphics, it is not a program itself (https://www.r-project.org/about.html); I suggest rephrasing this sentence to be more accurate, just a suggestion.

Line 271: “Our study showed important clinical IMPLICATIONS, because…”

Line 272: “…risk for GDM development (OR OTHER COMPLICATIONS), could promote…”

Author Response

In this article, the authors measured serum levels of HtrA1 in pregnant women using an ELISA approach, first in healthy women (n=20) and women with GDM (n=20) in the third trimester of pregnancy, and then in sera from women at the first trimester of pregnancy (n=238) from which 28 developed GDM. Results showed that HtrA1 serum levels were higher in women with GDM in the third trimester of pregnancy, and in women in the first trimester that developed GDM later in pregnancy. In addition, the results showed that the probability of developing GDM increased with increasing levels of HtrA1 and BMI values. These results suggest that HtrA1 could be an early predictive marker for GDM, and thus has clinical relevance.  Nevertheless, the increase of HtrA1 level in serum seems not to be specific for GDM, but also for other pregnancy complications, therefore the title could be misleading and might require a slight modification, for example “HtrA1 as early marker of pregnancy complications including gestational diabetes mellitus”

We agree with the reviewer, we modified the title in “HtrA1 in gestational diabetes mellitus: a possible biomarker?”

 Please provide a Table with complete information (baseline demographics -including ethnicity-, clinical history, habits, etc.) for each patient in a supplementary file. This information could be useful for the readers.

We added the information requested in Table S1 (for third trimester) and Table S2 (for first trimester)

 Lines 169-173: “Moreover, the median birth weight of children of women with GDM was 3.2 Kg (IQR: 2.8;3.4) while for normal pregnancies was 3.5 Kg (IQR: 3.3;3.6) proving that children of women with GDM had a statistically significant lower birth weight (P = 0.041), which was possibly due to the lower gestational age at delivery found in pregnancies complicated with GDM”. Some GDM babies are large for gestational age, the authors should report the weight of the baby according to gestational age, otherwise is not comparable to full term babies.

We thank the reviewer for this comment. We checked the data and we found a mistake in the copy/paste of the table in the journal template. In fact, there was no difference in gestational age at delivery (p=0.107) but the difference in birth weight is correct. However, it is well known that alteration of the glucose  metabolism  may be a cause of maternal inflammation with a possible damage on placental development. This can lead to lower neonatal weight. This has also been demonstrated in a previous study showing that, although not common, SGA can be a complication of GDM (PMID: 27269646). We added this comment into the manuscript to clarify the results (Lines 175-176)

 Even if HtrA1 serum levels were significantly higher in women with GDM, individual variability is high; it would be useful if the authors provide individual HtrA1 values of healthy and GDM pregnancies in a supplementary file.

We added the information requested in Table S1 (for third trimester) and Table S2 (for first trimester)

The authors discuss major strengths of the study, however they should also discuss limitations.

We added the limitations of the study.

 Minor points:

Line 63: “… inflammation IS one of the most important CAUSES OF insulin resistance…”

Corrected

Line 102: “… third trimester of GESTATION in normal…”

Corrected

Line 126: “A history…”

Corrected

Line 155: R is a programming language and environment for statistical computing and graphics, it is not a program itself (https://www.r-project.org/about.html); I suggest rephrasing this sentence to be more accurate, just a suggestion.

Thank you for the suggestion, we modified the sentence according to the reviewer suggestion

Line 271: “Our study showed important clinical IMPLICATIONS, because…”

Corrected

Line 272: “…risk for GDM development (OR OTHER COMPLICATIONS), could promote…”

Corrected

Round 2

Reviewer 2 Report

In this revised version of the manuscript the authors answered all my questions, I am satisfied with the answers and the effort is appreciated. I have only one minor observation, I suggest changing “gender” for “sex” in both supplementary Tables. Male and female are two sexes. Sex refers to the different biological and physiological characteristics of males and females (such as reproductive organs and sex hormones), while gender refers to the socially constructed characteristics of women and men (such as norms, roles, and relationships). I have no further comments.